# Determinants of Vaccine Hesitancy among African American and Black Individuals in the United States of America: A Systematic Literature Review

**DOI:** 10.3390/vaccines12030277

**Published:** 2024-03-07

**Authors:** Elena Savoia, Evelyn Masterson, David R. Olander, Emma Anderson, Anisa Mohamed Farah, Luca Pirrotta

**Affiliations:** 1Department of Biostatistics, Harvard T.H. Chan School of Public Health, 677 Huntington Avenue, Boston, MA 02115, USA; 2Emergency Preparedness Research, Evaluation & Practice Program, Division of Policy Translation & Leadership Development, Harvard T.H. Chan School of Public Health, 677 Huntington Avenue, Boston, MA 02115, USAdolander@hsph.harvard.edu (D.R.O.); emmalynandersonn@gmail.com (E.A.); aamf@hsph.harvard.edu (A.M.F.); luca.pirrotta@santannapisa.it (L.P.); 3Management and Healthcare Laboratory, Institute of Management and Department EMbeDS, Scuola Superiore Sant’Anna, Via S. Zeno 2, 56127 Pisa, Italy

**Keywords:** vaccine hesitancy, African American, Black, COVID-19 vaccine

## Abstract

Despite the crucial role the COVID-19 vaccine played in curbing the pandemic, a significant portion of Black and African American individuals expressed hesitancy toward being vaccinated. This review aimed to identify the determinants of COVID-19 vaccine hesitancy among Black and African American individuals in the U.S. The literature search was conducted in December 2022 according to the PRISMA criteria focusing on empirical studies. Data extraction methods, critical appraisal, and secondary thematic analysis were conducted on both quantitative and qualitative studies. Sixteen quantitative studies identified the key factors associated with vaccine hesitancy, such as confidence in vaccine effectiveness, safety, and trust in the healthcare system. Fourteen qualitative studies revealed major themes of mistrust, fear, and information needs, including historical mistrust, concerns about the vaccine development process, and contemporary institutional mistrust. The synthesis of quantitative and qualitative findings derived from this review provides a nuanced understanding of the determinants of vaccine hesitancy in Black and African American communities in the U.S., offering a foundation for the development of evidence-based interventions. Mistrust in the healthcare system, fear, and informational gaps on vaccine safety and effectiveness were identified as significant barriers to vaccination, demanding targeted interventions.

## 1. Introduction

Defined as “*a delay or refusal to accept vaccination despite its availability*” [1], vaccine hesitancy is a multifaceted issue—a complex phenomenon, which can vary in intensity and motivations across various segments of the population, impacting vaccine coverage and the effectiveness of public health interventions during infectious disease outbreaks [2]. As stated by the World Health Organization (WHO), “*vaccine hesitancy represents one of the top ten threats to global health*”, emerging as a substantial challenge to public health, as manifested during the COVID-19 pandemic [3]. During the pandemic, the spread of misinformation [4] and the insurgence of anti-vaccination movements [5,6] have contributed to an increase in global vaccine hesitancy despite the proven benefits of immunization practices. 

Despite mass vaccination being recognized as the most effective approach to curbing COVID-19 mortality rates, a considerable number of Americans remained hesitant toward receiving the vaccine [7,8]. Notably, disparities in vaccination rates have been observed across racial and ethnic groups [9], with particularly low rates reported among Black and African American individuals at the start of the vaccination campaign. Such disparities narrowed over time [9]. Vaccine attitudes have followed similar trends [10,11,12]. As shown in repeated cross-sectional surveys of Facebook users, differences in vaccine acceptance between Black and White respondents were more pronounced at the start of the vaccination campaign and decreased over time. In January 2021, 60% of Black respondents were hesitant about receiving the vaccine compared to 30% of White respondents. In May 2021, hesitancy among Black respondents was approximately 40%, marking approximately 20% difference with White respondents. As such, the level of vaccine acceptance among Black individuals improved dramatically over time, more than for any other race. However, the gap between races at the start of the pandemic may have caused a delay in curbing the spread of the disease, a situation which may repeat itself in the future and affect the response to future emergencies. As such, it is important to understand the reasons and learn from what occurred during the pandemic [13].

In Black communities, vaccine hesitancy is rooted in a troubling history of unethical medical experiments and persists today due to how this segment of the population still experiences discrimination, racism, mistreatment, and overall health inequalities [14]. In addition, Black communities are frequently affected by low income and low literacy levels, factors which limit access to crucial COVID-19 information, further impacting vaccination rates within this population [15].

An expanding body of literature is delving into the attitudes, perspectives, information sources, and communication preferences related to COVID-19 vaccine hesitancy within racial/ethnic communities. A recurring theme in this body of work is the historical distrust experienced by African American communities toward the healthcare system and governmental institutions [12,16]. Several studies have demonstrated how mistrust in government and pharmaceutical organizations has emerged as the main reason for COVID-19 vaccine hesitancy [8,17]. Furthermore, other studies have highlighted how vaccine hesitancy can stem from exposure to misinformation or misconceptions about the risks and benefits of vaccines [4,18]. In the context of the COVID-19 pandemic, concerns about the rushed development of vaccines and their potential side effects, as well as historical memory of unethical medical practices, have been recognized as prominent reasons for vaccine hesitancy among African American and Black individuals [14,17,19].

This literature review focuses on identifying the socio-demographic factors, attitudes, beliefs, and past experiences associated with COVID-19 vaccine hesitancy among the general population of Black and African American individuals in the U.S. With this literature review, we aim to contribute to the knowledge necessary to develop targeted, evidence-based approaches, which address the specific concerns and barriers faced by this population in accepting the COVID-19 vaccination.

Our review addressed the following question: What are the determinants of COVID-19 vaccine hesitancy among Black and African American individuals in the U.S.?

## 2. Materials and Methods

The review was guided by the Preferred Reporting Items for Systematic Reviews and Meta-Analyses (PRISMA) [20,21] statement and registered in PROSPERO (registration # CRD42022371229). 

Based on the Population, Intervention, Comparison, and Outcome (PICO) framework [22], we determined the population of interest to be Black and/or African American individuals in the U.S., the intervention to be the determinants leading to COVID-19 vaccine hesitancy, the comparison group to be other races, and the outcome to be COVID-19 vaccine hesitancy.

### 2.1. Search Strategy

We searched the literature through December 2022 for relevant peer-reviewed studies in English and Spanish in the following electronic databases: MEDLINE, Embase, PsycINFO, CINAHL, Cochrane Library, and Web of Science. Table A1 (see Appendix A) presents the keywords used in the search and the number of articles retrieved from each database. The study records were retrieved and exported using the software Covidence [23]. The review followed a four-step screening process. First, the Covidence reference manager software was used to combine the articles and eliminate any duplicates. Second, the articles were screened for relevance by reading the title and abstract. Third, we ensured that articles were available in full text through the Harvard library and met the inclusion and exclusion criteria, and, after a full text review, we extracted the content based on pre-set criteria. Finally, the studies were assessed for their quality and the findings summarized using qualitative approaches. Figure 1 shows the flow of screening the articles. 

### 2.2. Inclusion/Exclusion Criteria

We applied a series of inclusion and exclusion criteria during the screening and evaluation of articles. The studies included were those which (a) addressed COVID-19 vaccine hesitancy/acceptance among Black and African American individuals, regardless of how this construct was measured, (b) were conducted in the U.S., (c) analyzed socio-demographic and other factors associated with COVID-19 vaccine hesitancy/acceptance, and (d) were based on a primary data peer-reviewed empirical study. Articles for which the full text was not available through our library or inter-library services were excluded. We also excluded articles, which described vaccine hesitancy rates by racial/ethnic group without providing information on the determinants of vaccine hesitancy/acceptance or articles focusing exclusively on Black and African American individuals with unique characteristics based on their job category or disability status, not reflective of the general population. We excluded one article written by our research team to avoid biased judgement. 

### 2.3. Data Extraction

Guided by the literature review question and inclusion criteria, a standardized form and data extraction Excel document were developed with pre-identified criteria to extract and categorize the articles based on the following elements: (a) study design, (b) sample characteristics (i.e., demographics), (c) geographic area, (d) data collection method (e.g., interviews, surveys, focus groups), (e) outcome measure (how vaccine hesitancy/acceptance was defined and measured), and (f) results. Two members of the team independently reviewed the articles and extracted the information. Subsequently, they met to discuss discrepancies in the categorization process, and a third team member was consulted to solve any final disagreement.

### 2.4. Critical Appraisal

“*Critical appraisal (CA) is used to systematically assess the quality of research papers and to judge the reliability of the study being presented in the paper. CA also helps in assessing the worth and relevance of the study*” [24]. We adopted existing CA tools based on the study design to assess the quality of each study. More specifically, we used the Critical Appraisal Skills Programme (CASP) [25] checklist tool for qualitative studies and the BMJ appraisal tool for cross-sectional studies [26]. Two members of the team independently appraised each study and met to discuss and solve discrepancies. More specifically, the qualitative and quantitative–longitudinal studies were graded based on the appraisal criteria outlined by CASP [25]. For each criterion, a score of 1 was given when the criteria were met, and a score of 0 was given when the criteria were not met or there was not enough information to assess it. The cross-sectional studies were graded on a 20-point scale compiling the appraisal criteria outlined by the BMJ checklist [26]. For each criterion, a score of 1 was given when the criteria were met, and a score of 0 was given when the criteria were not met or there was not enough information to assess it. 

### 2.5. Synthesis Procedures

#### 2.5.1. Analysis of the Quantitative Studies

For each study, two team members extracted the list of variables, which were analyzed as potential determinants of vaccine hesitancy/acceptance, and coded the results based on the variable being “*associated with the outcome*” or “*not associated with the outcome*”, regardless of the statistical method used to analyze the association between the variables. A third reviewer was consulted to solve discrepancies in the coding. The results across studies were summarized by variable in narrative form because pooling data through quantitative methods was not possible due to the limited number of studies and variation in how the outcome was defined, in addition to the statistical techniques used to study the association between variables. A synthesis was provided only for variables, which had been analyzed by a minimum of three studies. An analysis of the methodological limitations encountered during the assessment of studies is provided as well.

#### 2.5.2. Analysis of the Qualitative Studies

Each qualitative study already included the results of a thematic analysis from its primary data obtained through focus groups and/or interviews. As such, our team conducted a secondary thematic analysis across studies using a deductive approach, aggregating results with similar themes. More specifically, two team members assigned codes to the results of each study and, through an iterative comparison, aggregated the themes to generate similar or larger constructs/themes and sub-categories/sub-themes. A third researcher was consulted to review the results and validate the interpretation of the themes. 

## 3. Results

We extracted 30 studies, 16 of which used quantitative methods (14 with a cross-sectional study design, 2 with a longitudinal study design) and 14 used qualitative methods (focus groups and interviews). 

### 3.1. Cross-Sectional and Longitudinal Studies (n = 16)

Characteristics of the Studies and Critical Appraisal

We identified 14 cross-sectional studies, all using online surveys. The average quality score was 13.8 out of 20. The characteristics of the studies are provided in Table A2 (see Appendix A). None of the papers provided a justification for their sample size; only one paper provided information on non-respondents to the survey and only two examined potential bias in the interpretation of the results. The two longitudinal studies, Wagner et al. [27] and Padamsee et al. [28], had an average CASP score of 9.5. It is worth noting that Wagner et al.’s [27] results used for this analysis are based on one cross-sectional sample within the longitudinal study. 

Below, we provide a summary of results based on the variables identified as potential determinants of vaccine hesitancy/acceptance among the 15 cross-sectional studies and 2 longitudinal studies. The *n* corresponds to the number of studies analyzing a specific variable/determinant. Figure 2 provides a visual representation of the number of studies analyzing each variable. The variables were classified based on the WHO’s 3C model of vaccine hesitancy [1].

#### 3.1.1. Age (*n* = 7)

Most studies (five out of seven) found a positive association between age and intent to get vaccinated. Cunningham-Erves et al. [29] found that individuals who had a higher intent to get vaccinated were older, on average. King et al. [13] studied a sample of Facebook users. They found differences in hesitancy by age to be more pronounced in Black individuals compared to Asian and White individuals. Younger adults, 18–24 years old, reported greater hesitancy compared to adults 75 years old. Reinhart et al. [8] determined that, for Black respondents, age was the only demographic variable positively associated with vaccine acceptance, showing older respondents as being more accepting of the vaccine than younger respondents. Willis et al. [30] found that, among Black adults in Arkansas, those reporting higher levels of COVID-19 vaccine hesitancy were younger, on average. Sharma et al. [31] showed that younger Black respondents were more hesitant about getting the vaccine. Contrary to these studies’ findings, Minaya et al. [32] found that age was not associated with intent to get vaccinated for Black individuals. Similarly, Williamson et al. [33] did not find any direct or indirect association between age and vaccine intentions in this population. 

#### 3.1.2. Gender (*n* = 8)

There are no consistent results regarding the association between gender and intent to get vaccinated. Cunningham-Erves et al. [29] found females to be less likely to intend to get vaccinated than males. Similarly, Ongubajo et al. [34] found women more commonly reported an intention to delay getting the COVID-19 vaccine than men. On the contrary, Wagner et al. [27] showed that, among Black non-Hispanic (NH) Detroiters, the odds of vaccination were higher among females than among males, while studies conducted by Bogart et al. [12], Willis et al. [30], Sharma et al. [31], and Minaya et al. [32] did not find an association between gender and vaccine acceptance. Interestingly, Reinhart et al. [8] showed an indirect and positive effect of female gender on vaccine acceptance through trust in institutions and physicians.

#### 3.1.3. Education Attainment (*n* = 5)

There are no consistent results regarding the association between education attainment and intent to get vaccinated. Cunningham-Erves et al. [29] found education attainment to be associated with vaccine intent in Black women. Those with less than a high school diploma and those with a college degree were less likely to have a lower intent compared to the middle category having a high school diploma. Wagner et al. [27] showed that among NH Black Detroiters, the odds of vaccination were higher in those with a college education compared to those with less schooling. On the contrary, three studies conducted by Reinhart et al. [8], Willis et al. [29], and Minaya et al. [32] did not find an association between education attainment and vaccination intent. 

#### 3.1.4. Income (*n* = 3)

Most studies (two out of three) show a positive association between income level and vaccine acceptance. Wagner et al. [27] found that, among NH Black Detroiters, the odds of vaccination were higher in those with an income of at least USD 50,000 vs. those with an income of less than USD 50,000 and that the interactions between race and income were statistically significant. Their interaction analysis revealed more income-based disparities among NH Black Detroiters than among other races or ethnicities. They also found that NH Black individuals with a higher income were more likely to intend to get vaccinated relative to those with a lower income. Williamson et al. [33] found that income was significantly related to COVID-19 vaccine intentions. On the contrary, Reinhart et al. [8] showed that, for the Black group, the standardized indirect effects of income on vaccine acceptance were not significant.

#### 3.1.5. Religiosity (*n* = 3)

Most studies (two out of three) show a positive association between religiosity, Christianity, and vaccine acceptance. Reinhart et al. [8] showed a standardized indirect effect of being Christian on vaccine acceptance. Sharma et al. [31] showed that religion, other than Christianity and Atheism, in the African American community was associated with higher vaccine hesitancy. On the contrary, Cunningham-Erves et al. [29] found that religiosity was negatively associated with COVID-19 vaccine uptake in Southeastern U.S.

#### 3.1.6. Political Affiliation (*n* = 3)

Most studies (two out of three) did not find an association between political party affiliation and vaccine acceptance. Reinhart et al. [8] found an indirect association between being affiliated with the Democratic Party and vaccine acceptance. On the contrary, Sharma et al. [31] did not find significant differences among vaccine-hesitant Black individuals based on political affiliation. Similarly, Minaya et al. [32] found that Democratic Party affiliation was not associated with intent to get vaccinated for Black individuals. 

#### 3.1.7. Confidence/Trust in Vaccine Effectiveness/Safety (*n* = 5)

All five studies demonstrated a positive association between beliefs about the safety and effectiveness of the vaccine and vaccine acceptance. Cunningham-Erves et al. [29] found that in Southeastern U.S., Black men and women’s confidence in COVID-19 vaccine effectiveness and safety was strongly associated with COVID-19 vaccination intent. McClaran et al. [35] found that Black participants who believed the vaccine is ineffective had less trust in the vaccination and less coping appraisal (perceived response efficacy) compared to those who did not mention such beliefs. Bogart et al. [12] reported that Black survey participants who held stronger mistrust in the vaccine itself were more likely to say that they would not get vaccinated. Taylor et al. [36] surveyed Black residents in Southeast Michigan and found that those who believed vaccines to be safe and effective were less hesitant to obtain the COVID-19 vaccine. Using repeated surveys, Padamsee et al. [28] found that the intention to get vaccinated improved over time for Black respondents more than for White respondents due to changes in beliefs about the vaccine’s safety and effectiveness and the necessity to get vaccinated. However, Padamsee et al. [28] found no evidence that the association between vaccination intent and either protection beliefs or safety and effectiveness beliefs was contingent on race. 

#### 3.1.8. Mistrust/Trust in the Health System and Providers (*n* = 6)

All six studies analyzing various forms of trust (e.g., in the healthcare system, in healthcare providers, in non-discriminatory practices) showed a positive association between such sentiment and vaccine acceptance. Prior to the rollout of the vaccine, Thomson et al. [37] found that, in Michigan, Black survey participants’ rejection of vaccine uptake (compared to White participants’) was partially mediated by medical mistrust. Minaya et al. [32] found that, across the U.S., ethnicity-based and general mistrust in the healthcare system was associated with lower intent to get vaccinated. Nguyen et al. [38] reported on Black hospitalized patients in Baltimore, Maryland, finding that distrustful patients were less likely to be willing to accept the vaccine when available than patients who trusted doctors and the healthcare system. Williamson et al. [33] found that, across the U.S., Black individuals’ trust in healthcare providers was associated with vaccination intentions, while mistrust of providers had an indirect effect on vaccination intentions through vaccine concerns. Wagner et al. [27] studied Black Detroiters and found that “*23% of the differences in vaccine uptake by race could be eliminated if these individuals were to have levels of trust in healthcare providers similar to those among white Detroiters*”. Similarly, Reinhart et al. [8] showed that trust in institutions, trust in physicians, and trust in non-discrimination significantly predicted vaccine acceptance among Black individuals. 

#### 3.1.9. Methodological Limitations

The studies we examined presented methodological limitations, which did not allow us to fully address the literature search question. The first limitation is inherent to the study design: cross-sectional studies limit our ability to determine causal relationships between variables. Some studies did not include confounding variables, further limiting the analysis of associations between potential determinants and vaccine hesitancy, and, when confounders were included, we found wide heterogeneity in the type of variables being investigated, limiting our ability to aggregate results. Most studies did not make comparisons between races; as such, we do not know the extent of the findings being specific to this population. 

Aggregation of results was also challenging due to the diverse geography and timing of when data were gathered across studies. Specific response issues were experienced at different points in time during the pandemic, and each state developed its own vaccination campaign strategy. In addition, there are differences in resources and overall public health infrastructure across states, which may affect how people feel about vaccines and how they access the facilities where vaccines are being distributed. In terms of sampling strategies, specific methodological issues were encountered as well because most studies were based on convenience samples. Furthermore, in most cases, it is not possible to extrapolate the results to the general population due to the sampling frame as well as the recruitment techniques frequently being based on specific online strategies (e.g., sampling Facebook users only). 

### 3.2. Qualitative Studies (n = 14)

Characteristics of the Studies and Critical Appraisal

Fourteen studies were identified as using qualitative methods and focused on the determinants of vaccine hesitancy/acceptance in minority groups, including African American and Black (AAB) individuals. Nine of the fourteen studies used focus groups for a total of 93 focus groups with 558 participants overall. Looking at participation among minority groups, 83 focus groups included AAB individuals for a total of 301 AAB focus group participants. Seven of the fourteen studies included interviews for a total of 222 subjects being interviewed of whom 169 were AAB individuals. It is worth noting that, when the study included participants from multiple minority groups, sometimes, the authors did not report the results by race but rather in an aggregate form. To the extent possible, when we analyzed and summarized the results of these studies, we focused only on what the authors explicitly attributed as issues related to vaccine hesitancy in Black and African American individuals. The distribution of participants in the focus groups and interviews as well as their racial/ethnic composition, sample characteristics, and specific results are provided in Table A3 (see Appendix A). Interestingly, only one study distinguished between African immigrants and African Americans in their sample [39].

All 14 studies received an overall score greater than 7 on a scale ranging from 0 (low quality) to 9 (high quality). However, many of the studies (9 out of 14) did not meet the CASP criteria related to the following question: “*Has the relationship between the researcher and participants been adequately considered?*”. Few authors critically examined their own role, potential bias, and influence during formulation of the research questions, data collection, and recruitment. 

Thematic Analysis

Figure 3 represents the themes and sub-themes related to African American and Black individuals’ COVID-19 vaccine hesitancy derived from the secondary analysis of the primary qualitative studies. The three major themes (Mistrust, Fear, and Information Needs) were represented in all 14 studies. In addition, all studies suggested interventions and solutions to reduce vaccine hesitancy, also summarized below. 

#### 3.2.1. Mistrust

All 14 studies [10,15,17,18,39,40,41,42,43,44,45,46,47,48] included in this analysis mentioned the construct of mistrust as a reason for vaccine hesitancy among African American and Black individuals. The following sub-themes were the most frequent types of mistrust mentioned in the literature: (a) historical mistrust, (b) mistrust of the vaccine development process, and (c) contemporary mistrust.

Historical Mistrust

Historical mistrust refers to the history of abusive and unethical research practices conducted on African Americans and was mentioned in 12 of the 14 studies. The Tuskegee Syphilis Study [10,18,40,41], the Henrietta Lacks case [39], and terms such as “guinea pig” [42] were used by focus group participants and interviewees to describe how African American and Black people were abused in medical research. To improve vaccination uptake, most studies emphasized the importance of acknowledging historical malpractice and expressing historical empathy in communication efforts to overcome judgment and appreciate the motives which may lead to COVID-19 vaccine hesitancy.

Mistrust of the Vaccine Development Process

Mistrust of the vaccine development process refers to challenges in understanding and trusting the technology and strategies used to rapidly approve the vaccine. Six of the fourteen studies specifically mentioned mistrust in this process [17,39,42,43,44], leading study participants to question the efficacy of the vaccine and its potentially unforeseen side effects [42]. In addition, two studies did not directly mention the term “*vaccine development*” but alluded to it by reporting that study participants had concerns about the speed of the process to create the vaccine [10,45]. Participants from three studies questioned whether the vaccine had been sufficiently tested [17,39,41]. According to one study, better educational outreach about the vaccine development process before the rollout of the vaccine could have helped mitigate this sentiment of vaccine hesitancy [43].

Contemporary Mistrust

All 14 studies mentioned some form of mistrust in government, politicians, and the medical establishment as a reason for COVID-19 vaccine hesitancy among African American and Black individuals [10,15,17,18,39,40,41,42,43,44,45,46,48]. When asked about information sources which people do not trust, study participants mentioned politicians and the government [42]. Many described structural racism [10], continued acts of injustice [43], neglect [48], and lower quality healthcare received by African American and Black individuals [44,48] as a reason not to trust the government and not to get vaccinated. Participants also expressed concerns regarding conflicting guidance from the government, which led to poor confidence in the overall COVID-19 response [41]. Participants in one study explicitly mentioned mistrust in pharmaceutical companies because such companies are perceived as being driven by profit rather than public interest [48]. Six studies cited inequalities and social injustice experienced in contemporary public health crises, such as the water crisis in Flint, Michigan [45], as a factor associated with a lack of trust in the COVID-19 vaccine [10,15,17,44,45,48].

#### 3.2.2. Fear of the COVID-19 Vaccine

All 14 studies included in this analysis mentioned issues related to the construct of fear as a reason for vaccine hesitancy among African American and Black individuals. The following sub-themes were the most frequent types of fear mentioned in the literature: (a) fear of unknown side effects and the vaccine being unsafe, (b) fear of being exposed to SARS-CoV-2 by the vaccine itself, and (c) fear of inequitable treatment or of being an object of experimentation. 

Fear of Unknown Side Effects and the Vaccine being Unsafe

Nine of the fourteen studies mentioned unknown potential side effects—both in the long and short term—as a reason for fear of the COVID-19 vaccine [10,17,41,42,43,46,47]. Of these nine studies, one did not specify whether the fear of side effects was expressed only by African American and Black individuals or whether it was common among other racial and ethnic groups included in the study sample (e.g., Latinx) [47]. Many African American/Black participants in these studies expressed a “wait and see” attitude [47], postponing vaccination due to uncertainty and concerns about potential side effects of the vaccine [10]. Ongoing negative media coverage on the side effects of the Johnson & Johnson vaccine impacted how participants viewed vaccination [43]. Similarly, six studies also discussed safety concerns [15,39,40,41,43,45], and participants in one study expressed their concern that rapid development of the vaccine may not adhere to the usual strict protocols [15]. One study also stated that they would be reassured by knowing that clinical trials for the vaccine included population samples representing their own racial and ethnic community, age group, and health conditions [45].

Fear of being Exposed to SARS-CoV-2 by the Vaccine Itself

Three studies included participants who expressed fear of being exposed to SARS-CoV-2 during vaccination [15,42,43]. Focus groups discussed a debilitating fear of contracting the virus, particularly due to its deadly nature and knowing people who had died of the disease. Fear of exposure to the virus led older individuals to avoid doctors’ offices where they could get vaccinated [42]. Preferences for the location of the vaccination clinics included familiar or local sites instead of a mass vaccination site with long lines or crowds, which would increase the risk of exposure to the virus [45].

Fear of Inequitable Treatment or of being an Object of Experimentation

One study mentioned fear of inequitable or differential treatment [45] and two similarly mentioned fear of experimentation or of being treated as a “guinea pig” [17,47]. Participants in one study feared receiving differential treatment in access and quality of the vaccine distributed to the population, projecting that wealthy White communities would receive higher quality vaccines and better vaccine management during the vaccination campaign [45].

#### 3.2.3. Information Needs

All 14 studies included in this analysis talked about a lack of clear, complete, and reliable information as a reason for vaccine hesitancy among African American and Black individuals [10,15,17,18,39,40,41,42,43,44,45,46,48]. Of these, one study did cite misinformation as a determinant of vaccine hesitancy, although it was unclear whether the determinant was applicable to African American and Black individuals included in the sample or only the Latinx/AAPI individuals cited directly in quotations [48]. Six studies specifically discussed the role of unclear or incomplete information in reinforcing vaccine hesitancy among AAB individuals [15,18,39,40,42,45]. However, once again, it was unclear in one study whether the results were applicable to the Black individuals included in the study or whether they were more generic to the broader demographics represented in the sample [45]. One study specifically emphasized the need for information on the composition of the vaccine, where people could receive the shot, and information about vaccine dosing and scheduling [39]. Two studies mentioned overwhelming issues of inconsistency in information from multiple sources, with poor explanations and lack of clarity making people feel confused as well as exhausted [18,40].

#### 3.2.4. Recommended Interventions Based on the Qualitative Studies

Twelve of the fourteen studies mentioned that community leaders and community-based organizations may play an important role in messaging and influence vaccine uptake [10,15,17,18,39,40,42,43,45,46,47]. Participants from one study agreed that what was needed to counter mistrust was information from trusted sources [42].

Religious leaders were mentioned as important messengers of vaccine information [17,18,39]. One study mentioned the importance of encouraging community health workers to talk to youth about the vaccine to increase willingness to get vaccinated [18]. Other interventions mentioned in the literature included hosting a community health day and disseminating information at local football/basketball games [18]. Seven of the fourteen studies highlighted that health practitioners’ recommendations in favor of the vaccination—without specifying the type of practitioners—could reduce vaccine hesitancy [15,17,18,39,41,44,45]. Participants discussed wanting to hear from medical professionals they could trust about the vaccine, and African American participants highlighted the importance of hearing from Black doctors [39]. Many also agreed that an open dialog with one’s doctor can contribute to more transparent sharing of information and an increased willingness to get vaccinated [18].

## 4. Discussion

The greater prevalence of pre-existing health conditions—such as obesity and cardiovascular diseases—within Black and African American communities, coupled with social and economic challenges experienced during the pandemic, has further underscored the difficulties which a public health system may face in achieving an equitable response during a crisis. Despite facing a greater burden of disease, vaccination rates among these communities have been low compared to White groups. U.S. states, which gathered information about race at vaccination sites, have revealed Black and African American population sub-groups exhibiting lower vaccination rates compared to non-Hispanic White individuals, with greater disparities at the start of the vaccination campaign [49]. 

As described by Roat C. et al. [50], several barriers have limited access to the vaccine for Black Americans, some of which are related to individuals’ feelings and experiences—such as lack of trust in the medical establishment and concerns about the safety of the vaccine—and others being due to systemic issues, such as limited access to healthcare services and other resources (e.g., transportation). As documented by Siegel M. et al. [51], structural racism certainly played a role in impacting the disparities in vaccination rates. The authors showed an association between structural racism with “differences in the magnitude of the observed racial disparities in COVID-19 vaccination”. 

To overcome the disparities in vaccination rates, some states have proactively developed vaccine distribution plans aiming to achieve equity in access to the vaccine based on the racial and ethnic composition of their population [52]. At the federal level, agencies such as the Centers for Disease Control and Prevention (CDC) and the Health Resources and Services Administration (HRSA) have allocated vaccines prioritizing community health centers, with the intent to reach the most vulnerable groups across states [53]. Despite extensive planning of the logistical efforts to achieve an equitable distribution and facilitate access to the vaccine for those most in need, local, state, and federal agencies have encountered challenges in addressing vaccine hesitancy, particularly among Black and African American individuals. While vaccine hesitancy transcends racial boundaries, various population sub-groups may harbor distinct reasons for refusing vaccination. This literature review aimed to identify the specific characteristics and reasons contributing to vaccine hesitancy in these population sub-groups. 

To achieve this goal, our initial examination of the literature focused on socio-demographic factors. This exploration revealed distinctions in vaccine acceptance among Black and African American individuals based on age and income, mirroring findings from the broader COVID-19 vaccine hesitancy literature [54]. Less clear was the association between gender, educational attainment, and vaccine acceptance, which also finds larger agreement in the general vaccine hesitancy literature [54].

It is noteworthy that, during the pandemic, a disproportionate number of elderly individuals were affected by the disease, along with a higher proportion of males compared to females. Consequently, it is anticipated that these socio-demographic groups may have displayed less interest in vaccination compared to segments of the population at higher risk of disease severity and death, irrespective of race [55,56].

Our review reveals that vaccine hesitancy in Black and African American individuals is primarily driven by concerns regarding the safety and effectiveness of the vaccination. This finding aligns with other studies focusing on the general population [57]. Specifically, these concerns are linked to apprehensions about vaccine side effects, uncertainty regarding the necessity of vaccination, and overall doubts about its effectiveness. This underscores the importance of future campaigns addressing the safety of pharmaceutical interventions and the need to provide clarity on what individuals can expect from the vaccination in terms of its effectiveness. Emphasis should be placed on effectively communicating the impact of the vaccine on disease severity vs. infection rates, as the communication strategies for COVID-19 vaccination in this regard have lacked in adequacy and comprehensiveness. The association between Christianity and vaccine acceptance found in some studies is certainly interesting and is a potential indicator of a history of partnerships between public health agencies and faith-based organizations for this faith group. 

A sentiment of trust, in its various forms—including trust in government, the healthcare system, and healthcare providers—has been identified as a significant factor contributing to vaccine hesitancy in the U.S. and worldwide [58,59,60]. The relationship between such sentiment and vaccine acceptance is similarly present among Black and African American individuals in the U.S., with an additional layer of complexity arising from a historical backdrop of unethical practices in the medical field and contemporary experiences of discrimination affecting this population’s sentiments of trust in the system and providers in charge of vaccination. 

Given the rapid advancements in biotechnology and modern vaccine production techniques, it is imperative for government agencies and pharmaceutical companies to invest in educational campaigns aimed at rebuilding trust in the medical and research establishments before the next emergency occurs. Rebuilding trust can be accomplished not solely through educational interventions, which demystify the technical facets of contemporary research methods, but also by guaranteeing fair access to pharmaceutical products and daily medical interventions. This is particularly pertinent in addressing a range of medical conditions, which impact vulnerable groups on a day-to-day basis. Public health initiatives during peacetime play a pivotal role in addressing health disparities in the U.S. and rebuilding trust in the public health system. This encompasses fostering confidence not only in healthcare providers but also in public health agencies and pharmaceutical companies.

The literature underscores the significance of customizing interventions to specifically address the distinct concerns and issues encountered by Black and African American individuals [11,39,61]. For instance, trusted messengers, such as healthcare providers and community leaders, including faith-based leaders, have been consistently identified as crucial influencers in the promotion of vaccine uptake within these communities [62]. While the articles we identified did not specifically address the role which exposure to misinformation may play in vaccine acceptance, a recent review suggests that people who have experienced discrimination and racism can be relatively likely to encounter medical misinformation and the challenges, which such low-quality information poses [63]. 

From a methodological point of view, our analysis allowed us to identify the opportunities to improve future research efforts in this field, starting with the need to identify valid and reliable measures of vaccine hesitancy, so that studies in this field use similar scales; developing longitudinal cohorts of respondents, so that causal relationships between variables can be determined; identifying sampling strategies, which include individuals with limited access to the internet; and selecting confounding variables beyond demographic factors, so as to factor potential barriers to accessing healthcare services and experience with such services in future interventions. 

Finally, recognizing and understanding the challenges, which Black individuals face in receiving proper healthcare during “peacetime”, as well as the daily obstacles impacting their wellbeing, is imperative for developing effective communication strategies and comprehensive responses to future emergencies. Communication, especially at the onset of a crisis, should be rooted in the principles of empathy, transparency, and accountability within the realm of public health practice. 

### Limitations

Despite a fairly large number of studies (*n* = 30) focusing on vaccine hesitancy among African American and Black individuals, there is large variation in the methodologies used to measure vaccine hesitancy and the factors being analyzed to determine what drives such hesitancy. For example, some studies focus on willingness to get vaccinated, while others focus on refusal of the vaccination; however, there is no agreed-upon measurement approach or validated scale to measure either of the constructs. As such, survey questions intended to measure these outcomes are developed in various forms, with no consistent and agreed-upon methods. Future research should focus on the development of validated scales. Furthermore, different statistical methodologies are used to test for the association between specific variables (e.g., socio-demographic factors) and vaccine acceptance, making it difficult to aggregate the results across studies. Qualitative studies are certainly the richest in terms of information on the specific reasons why African American and Black individuals show lower intent to get vaccinated compared to White individuals. However, in many cases, focus groups and interviews included several types of minority groups and merged African American with Latinx without providing specific findings based on race/ethnicity. In addition, most studies do not make a distinction between African American and Black individuals who were born in the United States from more recent African immigrants who might have a different experience with vaccination practices and trust in the healthcare system in the U.S. 

## 5. Conclusions

This synthesis of 16 quantitative and 14 qualitative studies provides a nuanced understanding of the determinants of COVID-19 vaccine hesitancy among Black and African American communities in the U.S., offering a foundation for the development of evidence-based interventions. Mistrust in the healthcare system, fear, and information gaps on vaccine safety and effectiveness were identified as significant barriers to vaccination, demanding targeted interventions. 

## Figures and Tables

**Figure 1 vaccines-12-00277-f001:**
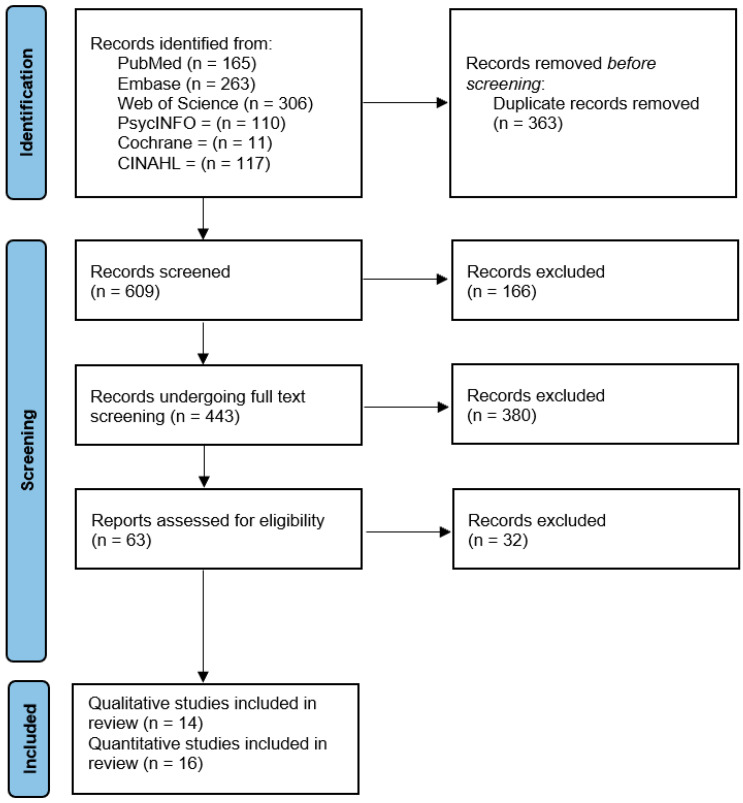
PRISMA statement flowchart [20].

**Figure 2 vaccines-12-00277-f002:**
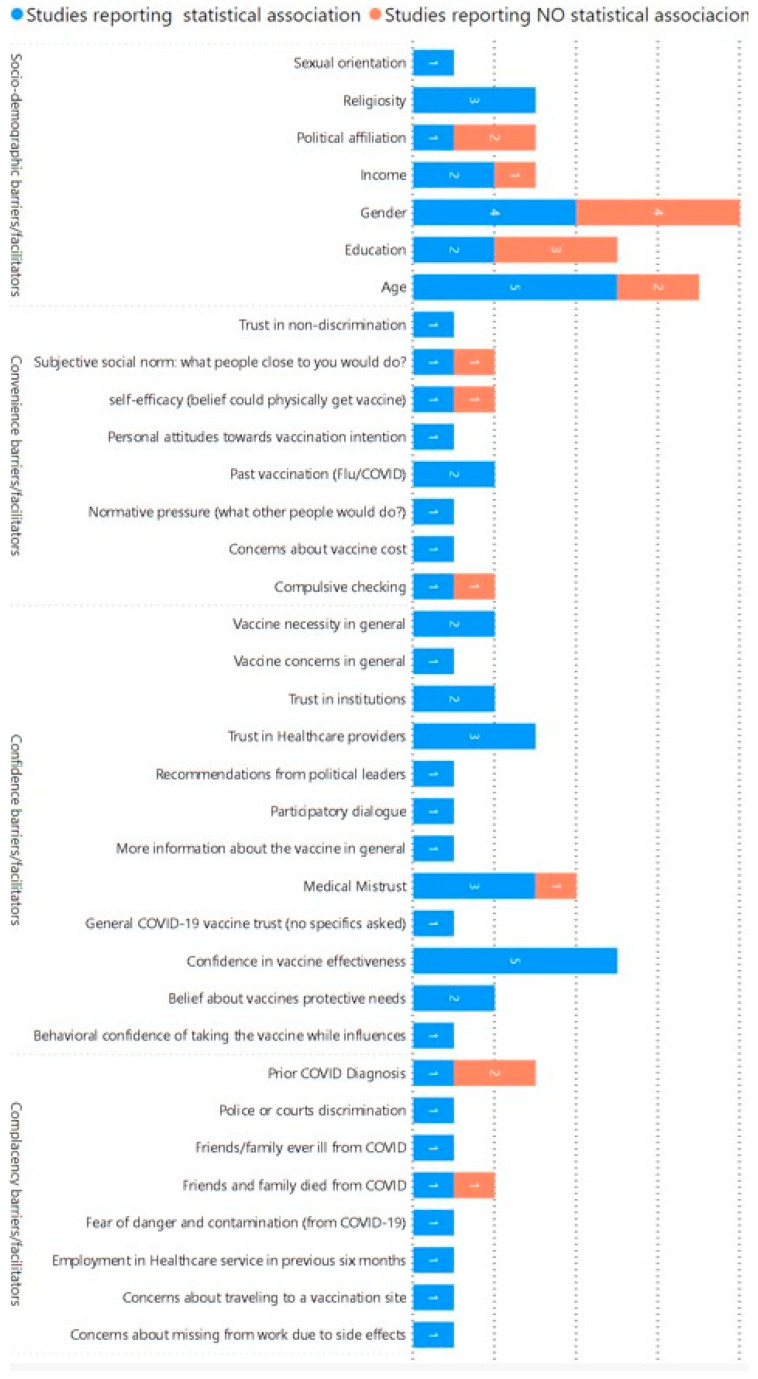
Distribution of variables examined by the quantitative studies (based on the 3C model [1]).

**Figure 3 vaccines-12-00277-f003:**
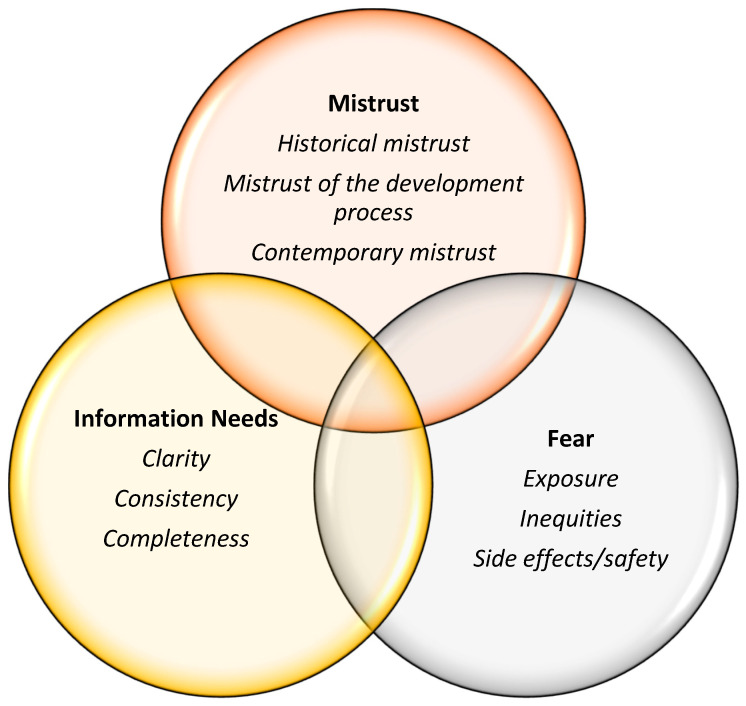
Themes and sub-themes emerging from the analysis of qualitative studies.

## Data Availability

No new data were created or analyzed in this study. Data sharing is not applicable to this article.

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
