# Peer review of "Determinants of Vaccine Hesitancy among African American and Black Individuals in the United States of America: A Systematic Literature Review"

_vaccines, 2024, doi:10.3390/vaccines12030277_

Round 1

Reviewer 1 Report

Comments and Suggestions for Authors

The authors have submitted an extensive consideration of relevant literature which suggests reasons why African Americans and Black individuals have demonstrated hesitation regarding vaccination. To provide the most useful contribution to the journal, however, the authors should go further to consider the nature of available evidence in this domain.

Health disparities between Black people and other groups pose a serious threat to our collective wellbeing and deserve scholarly attention. What is less clear is the extent to which those disparities are driven by personal openness to vaccination, per se, as opposed to factors such as structural barriers to health care. In a review paper such as this, the authors should summarize evidence for vaccination rate differences over time and then also summarize what we know about different perceptions among people.

At least some of the articles cited by the authors do not include data for group comparisons as much as they provide data relevant to describing a particular group. The authors do highlight some of the available data, e.g., comparing Black Detroit residents with white Detroit residents but throughout the paper it would be helpful if they could distinguish between data on associations versus evidence of causal relationships and in cases where causal evidence is lacking what evidence we need.

The authors also begin to cite the potential influence of misinformation and misperceptions. One recent review which might also be helpful to consider in this vein in an Annual Review of Public Health piece which suggests lack of access to high quality health information challenges communities which have historically faced health disparities but that lack of access is not a useful primary explanation for those disparities: https://www.annualreviews.org/doi/abs/10.1146/annurev-publhealth-071321-031118. 

If the authors can more clearly note the methods strengths and weaknesses of the papers they cite, including what is and isn't possible to conclude based on the typical study samples of articles cited, they will improve the paper. They already have started an important line of critique with their exploration of inconsistencies in how variables like trust are measured and so if they can further frame this paper as both a reporting of what we know and as a guide for future research that will be helpful. 

Author Response

please find attached the response in the pdf. 

Reviewer 2 Report

Comments and Suggestions for Authors

Elena Savoia et al discuss in their review article why black individuals in the United States have a higher hesitation to let themselves vaccinate than the other ethnic groups. 

The authors considered 972 articles from which they finally selected 16 with a quantitative study of this subject and 14 with a qualitative study. The authors determined as main determinants for the hesitation to get vaccinated mistrust in government and pharmaceutical organisations and misinformation about the risk and benefits of vaccines. 

The conclusions the authors draw are that emphasis should be placed on campaigns concentrating on effectiveness and side effects of vaccines brought forward by trusted mediators of this community. 

This article is a very good review of the investigations into the reasons why the black population in the United States tends to have a lower vaccination rate that other ethnicities. Already the process of selecting articles that will be investigated further is exemplary. The article is well structured and its contents easy to understand and well presented. The conclusions are perfectly supported by the presented data. 

I think this article can be published as it is. 

Author Response

please find attached the response in the pdf file. 

Reviewer 3 Report

Comments and Suggestions for Authors

For public health authorities, acceptance  of most vulnerable population groups is a key determinant  of vaccination effectiveness. This studies review provides usefull  qualitative information to better design strategic interventions against hesitancy in the selected population

The methodogy for the selection of publications is sound with clear exclusion criteria.

The great variation of methodologies used by these selected studies makes the analysis of results quite complex. 

I suggest that you put table 1 and 2 in annex. 

Unfortunately, it seems that studies reviewed have not provided analyses of associations of hesitation parameters ( multi factorial approach), links with access to vaccination facilities and co-morbidities conditions

I do not see how you could improve your work with such limitations;

However, from your extensive review, you are in a position to make some general appraisal of the methodology challenge and come with usefull conclusions for the research community. In this regard, you may draft a special section.

Author Response

please find attached the response in the pdf file

Round 2

Reviewer 1 Report

Comments and Suggestions for Authors

The authors have addressed reviewer comments and improved the manuscript. Their new discussion of misinformation and its relationship to structural disparities, though, needs revision. They note "a recent review suggests that people who have experienced discrimination and racism can be more susceptible to misinformation" and yet that is not what that review reports. That sentence would be more accurate if revised to "can be relatively likely to encounter medical misinformation and the challenges that such low-quality information poses." (The notion of inherent susceptibility differences doesn't really enjoy empirical support.) The authors should make that revision before publication. 

Author Response

We certainly agree with the reviewer and have replaced the original sentence with the one suggested. Thank you.